# Content of Health-Promoting Fatty Acids in Commercial Sheep, Cow and Goat Cheeses

**DOI:** 10.3390/foods11081116

**Published:** 2022-04-13

**Authors:** Arkadiusz Szterk, Karol Ofiara, Bartosz Strus, Ilkhom Abdullaev, Karolina Ferenc, Maria Sady, Sylwia Flis, Zdzisław Gajewski

**Affiliations:** 1Transfer of Science Sp. z o.o., Strzygłowska 15, 04-872 Warsaw, Poland; kofiara@transferofscience.com (K.O.); bstrus@transferofscience.com (B.S.); iabdullaev@transferofscience.com (I.A.); 2Center for Translational Medicine, Warsaw University of Life Sciences, Nowoursynowska 100, 02-797 Warsaw, Poland; karolina_ferenc@sggw.edu.pl (K.F.); maria_sady@sggw.edu.pl (M.S.); sylwia_flis@sggw.edu.pl (S.F.); zdzislaw_gajewski@sggw.edu.pl (Z.G.)

**Keywords:** medium-chain fatty acids, branched-chain fatty acid, vaccenic acid, conjugated dienes of linoleic acid, mold cheese

## Abstract

The study aimed to examine samples of different market original sheep cow and goat cheeses, in respect of the content and profile of FA with special emphasis on health-promoting FA. The content of fatty acids in the examined cheeses was highly differentiated and depended on the sort and type of cheese. The content of fatty acid groups in milk fat varied within the limits: SFA, 55.2–67.2%; SCSFA, 10.9–23.4%; BCFA, 1.6–2.9%; MUFA, 15.2–23.4%; PUFA, 1.9–4.3%; *trans*-MUFA, 1.8–6.0%; and CLA, 1.0–3.1%. From among the examined cheeses, the seasonal sheep cheeses (Oscypek) and mountain cow cheeses were characterized by the highest content of health-promoting fatty acids. The content of health-promoting fatty acids in the fat fraction of these cheeses was CLA 2.1–3.1%, *trans*-MUFA 3.5–6%, BCFA 2.7–2.9%, and SCSFA 12–18%.

## 1. Introduction

The milk fat of ruminants is considered one of the most complex naturally occurring fats. Researchers, by using chromatographic and spectroscopic techniques, identified about 400 FA with a different number of carbon atoms from 2 to 28. Milk fat has the following groups of FA characterized by containing even and odd numbers of carbon atoms with different degrees of saturation (saturated, mono-, and polyunsaturated) and *cis* and *trans* configurations, as well as straight and branched carbon chains [1]. The vast majority of these acids occur in a very small quantity (<0.1%). More than half of the FA of dairy products are saturated. The specific health effects of individual FA have been extensively studied. Stearic acid (18:0) does not seem to increase serum cholesterol concentration and is not atherogenic [2,3]. It would appear, accordingly, that some of the saturated FA in dairy products have neutral or even positive effects on health. In contrast to this, the saturated FA lauric-, myristic-(14:0), and palmitic (16:0) acid have LDL and HDL cholesterol-increasing properties [3]. High intake of these acids raises blood cholesterol levels [3], and diets rich in saturated fat have been regarded as contributing to the development of heart diseases, weight gain, and obesity [4]. Association between consumption of milk and milk products and serum total cholesterol, LDL cholesterol, and HDL cholesterol has been reported [5]. High cholesterol levels are a risk factor for coronary heart disease, with LDL cholesterol and a high ratio between LDL and HDL cholesterol enhancing the risk of coronary heart disease [6,7]. Several intervention studies have shown that diets containing low-fat dairy products have been associated with favourable changes in serum cholesterol [8,9,10]. However, dairy product fat consumption has been shown to have less pronounced effects on serum lipids than could be expected from the fat content [11,12].

Health-promoting fatty acids of the milk fat of ruminants include SCSFA, with 4 to 12 carbon atoms in the molecule—C4:0–C12:0). They reduce the risk of obesity and advantageously influence the energy balance of the human being [13]. This thesis is confirmed by many studies carried out on animal models in which the SCSFA have been found to indicate an effect in reducing body weight [14,15].

The effect of reduction of body weight caused by consumption of SCSFA was also confirmed in studies with the use of people. SCSFA also indicate an important role in antimicrobial activity.

It has been shown that lauric acid C12:0 and capric acid C10:0 are characterized by bactericidal activity against a number of pathogens present in food, including *Listeria monocytogenes*, *Clostridium perfringens*, *Escherichia coli*, *Salmonella enteritidis,* and *Campylobacter jejuni* [13].

The antibacterial activity of lauric acid against Staphylococcus aureus was indicated in a dose of 400 mg mL^−1^ [16].

The discovery of the inactivating action of SCSFA acids (resulting from the hydrolysis of milk fat with the involvement of human lipase) against *Helicobacter pylori* was a breakthrough [17].

Another valuable group of health-promoting fatty acids, from a nutritional point of view, are monounsaturated fatty acids, especially vaccenic acid.

Currently, much is known about the effect on lipid metabolism of the principal FA in the human diet. Based on this knowledge, dietary recommendations have been made for the population with the aim of reducing cardiovascular risk. Given that the most noteworthy effect is that Saturated FA increases plasma levels of total cholesterol and LDL cholesterol, it is widely accepted that a healthy diet should contain a limited amount of this nutrient. The effect of a high intake of MUFA results in a wide range of health benefits beyond cholesterol, raising great interest in the possible preventive effects of this type of diet on cardiovascular risk. MUFA-enriched diets reduce the requirement for insulin and decrease the plasma concentration of glucose and insulin in type 2 diabetic patients, compared with the effect of high SFA and low-fat, high cholesterol diets [18]. Certain data show that this dietary model could have a hypotensive effect, similar to that observed with the intake of other unsaturated fat-enriched diets. Furthermore, substantial evidence suggests that oleic-enriched LDL are more resistant to oxidative modifications and dietary MUFA could influence different components and functions related to the endothelium [18]. One of the most important FA found in dairy products fat is vaccenic acid. This acid belongs to the MUFA group. The hypolipidemic action of C18 11 and *t*11 was indicated in studies on animals. Bassett et al. (2010) [19] and Gómez-Cortés et al. (2018) [20] indicated the reduction of the concentration of triglycerides and LDL in the organism of mice consuming feed containing 15% of butter with a 1.5% share of vaccenic acid. In recent studies by Jacome-Sosa et al., (2014) [21], the authors indicated new mechanisms by means of which vaccenic acid may affect the metabolism of lipids, energy use, and fat tissue distribution in the body. A very important function of vaccenic acid, being the subject of numerous scientific studies, is the inhibition of the development of tumour cells [20,22].

A characteristic feature of the fat milk of ruminants is the occurrence of health-promoting conjugated diene of linoleic acid called CLA. This name means a mixture of positional and geometric isomers of octadecadienoic acid, C18:2, with two conjugated double bonds which, when isolated, constitute one single bond. These isomers most often occur at positions 8 and 10, 9 and 10, 10 and 12, and 11 and 13.

The ssomers of CLA effectively inhibit the proliferation of tumour cells of mamma, melanoma, colon, prostate, ovarian, and leukaemia [23,24]. CLA has also shown beneficial effects in reducing the risk of atherosclerosis [25]. CLA isomers play a very important role in the synthesis of eicosanoids whereby they have a direct influence on the functioning of the human immune system.

CLA has been shown to modulate the immune system and prevent immunodeficiency in animals [26,27,28,29,30]. The last very valuable health-promoting group of fatty acids occurring in the milk fat of ruminants are BCFA. The main BCFA of ruminant milk fat are isomers of tetradecanoic, pentadekanoic, hexadecanoic, and heptadecanoic acid [31,32], and its content mainly depends on dietary factors as in the case of the other groups of fatty acids discussed earlier [31].

An important property of BCFA is their anti-tumor activity. BCFA, both in vitro and in vivo conditions, effectively inhibit the growth of cancer cells in many organs (e.g., liver, lung, pancreas, prostate, and mammary gland) [33]. In recent studies, it has been shown that KT from the branched group (BCFA) are components of many tissues and organism fluids of the human organism/body: the gastrointestinal tract of infants, human milk, skin lipids, amniotic fluid, and meconium (intestinal contents of neonates) and vernix caseosa (white substance covering the body of newborns during birth) [34].

Depending on the quality of the starting raw material (milk), it is possible to shape the fatty acid composition in cheese [1,35]. Pintus et al. (2013) [25] published the results of many years of research conducted on a large population of people on the possibility of using sheep cheese as a functional food to reduce LDL in the human body.

Sheep cheese containing 11% of SCFA, 6% of vaccenic acid, and 2.8% of CLA in its fat fraction significantly decreased the level of LDL in people with cholesterolemia and the concentration of triacylglycerols without observable side effects by 7%. Having this in mind, the study of fatty acid content in different grades of cheese is very important, especially in terms of the content of health-promoting fatty acids such as SCFA, BCFA, *trans*-MUFA (vaccenic acid), and CLA.

On the other hand, dairy foods have been viewed with suspicion by many because dairy fats contain saturated fatty acids and cholesterol. It has been thought, particularly by consumers, that dairy fats may increase the risk of coronary heart disease because of the contribution they make to total saturated fat intake. However, dairy fats contain other lipid bioactives (e.g., omega-3 fatty acids, gangliosides, conjugated linoleic acid) that may counteract the effect of saturates in a well-balanced diet. Surprisingly, there have been few studies that have addressed this issue. In our study, we focused on analysing pro-health fatty acids, which can be found in various cheese samples. Based on our own results and scientific data in terms of pro-health fatty acids, we proved that some cheese can be qualified as a functional food.

The aim of the study was to examine 320 different samples of the original sheep, cow, and goat cheeses, both mold and yellow ones, in terms of the content of fatty acids, with special emphasis on health-promoting fatty acids.

The aim and the novelty of the study were also to check which of the examined cheeses could fulfil the criteria of functional food regulating the concentration and profile of the lipid fraction in the human body (concentration of LDL cholesterol and triacylglycerol) based on the results of the studies by Pintus et al. (2013) [25] and Wilms at al. (2022) [36], as well as Commission Regulation (EU) No 432/2012 of 16 May 2012 establishing a list of permitted health claims made on foods other than those referring to the reduction of disease risk, and to children’s development and health.

## 2. Materials and Methods

### 2.1. Materials

In total, *n* = 320 samples of cheese have been examined, taking into account the repetitions. Table 1 includes additional information regarding the examined cheeses, such as type of milk from which they were obtained, type of cheese (mold cheese and yellow cheeses), and classification of cheese in terms of their texture.

### 2.2. Chemicals

Supelco 37 Component FAME (mix certified reference material, TraceCERT^®^, in dichloromethane, CRM47885 SUPELCO, Poland, Poznań), linoleic acid, conjugated methyl ester (MERCK cat. number: O5632, Poland Poznań), linoleic acid methyl ester mix (*cis*/*trans*, certified reference material, 10 mg/mL in methylene chloride, cat. number: CRM47791, SUPELCO, Poland, Poznań), linoleic acid methyl ester mix (*cis*/*trans*, certified reference material, 10 mg/mL in methylene chloride, cat. number: CRM47791, SUPELCO, Poland, Poznań), linolenic acid methyl ester isomer mix (cat. number: L6031 MERCK, Poland, Poznań), linolenic acid methyl ester isomer mix (certified reference material, 10 mg mL^−1^ in dichloromethane, cat. number: CRM47792, SUPELCO, Poland, Poznań), sodium methoxide reagent grade, 95% (cat. number: 164992, MERCK, Poland Poznań), methanol ultrapure an hydrous, 99.8% (cat. number: 322415, MERCK, Poland Poznań), disodium hydrogen citrate sesquihydrate ReagentPlus^®^, 99% (cat. number: 359084, MERCK, Poland, Poznań), sodium chloride BioXtra, ≥99.5% (cat. number: S7653, MERCK, Poland, Poznań), Methyl undecanoate (C11:0 FAME), purity ≥ 99% mass fraction (first internal standard, cat. number: 94118, MERCK, Poland, Poznań), tritridecanoin (C13:0 TAG), purity ≥ 99 % mass fraction (second internal standard, cat. number: T3882, MERCK, Poland, Poznań), n-hexane for gas chromatography (cat. number: 1.00795, MERCK, Poland, Poznań).

*Cis*/*trans* FAME isomer standard mixture 5.5 mg mL^−1^ in 5 mL of methylene chloride, bacterial acid methyl esters CP mixture in methyl hexanoate (branched-chain fatty acids mixture standards, Catalogue No: 1114) was bought in Matreya LLC Lipids and Biochemicals, State College, PA USA.

### 2.3. Fat Determination

Fat was extracted from cheeses using the method of Röse-Gottlieb AOAC 905.02 [37,38]. For extractions of fat, three samples of 3 g of each cheese were taken.

### 2.4. Fatty Acid (FA) Analysis

FA methyl esters were prepared and calculated according to AOAC Official Method 2012.13 [39] protocol and analysed via gas chromatography [39] using Bruker Daltonics gas chromatograph (Billerica, USA) model 450-GC equipped with a RESTEKRT 2560 (100 m × 0.25 mm × 0.2 mm) capillary column and an FID. The injection volume of the samples was 2 µL. The temperature program used was the following: initial temperature 50 °C for 2 min, then increased to 175 °C at 3 °C/min and maintained for 20 min, then increased to 240 °C at 6 °C/min and maintained for 25 min. Peak identification was accomplished through the analysis of analytical standards. Fatty acid methyl esters (especially BCFA, *cis*/*trans* isomers C18:1, C18:2 and C18:3) were confirmed using a gas chromatograph coupled with a mass spectrometer Shimadzu (Kyoto, Japan) model GCMS-TQ8040 using exactly the same capillary column and temperature gradient program and injection of the sample. The identification of unknown fatty acids has also been supported by literature data [40,41].

### 2.5. Statistical Analysis

The obtained results were statistically analysed using Statistica 10 software (StatSoft Inc. 2012; Kraków, Poland). Mean values and standard deviations (SD) were calculated, the *n* = 8 for each cheese. The following statistical methods were employed: one way ANOVA, hierarchical and non-hierarchical analysis, and classification tree analysis.

The hierarchical analysis involved the use of the Ward agglomeration method, and the squared Euclidean distance was employed as the measure of distance. Criteria for the number of clusters were determined based on the course of the agglomeration process, and cluster significance was determined using a one-way analysis of variance at α = 0.05 and the LSD test. The chart showing a dendrogram includes the course of the bonding distance in relation to the occurrences based on which the number of clusters was determined in the place where the element aggregation increased most rapidly (bonding distance).

Hierarchical analysis was the basis for non-hierarchical analysis using a certain number of significant clusters. This analysis was essential for the classification tree analysis, which aimed at establishing a hierarchy of importance of different variables in determining the fatty acids in studied cheese samples.

## 3. Results

The results of fat content in the tested cheeses are presented in Table 1. The fat content significantly (*p* < 0.05) depended on the type of cheese and ranged from 18.0% for Le Fleuret to 33.6% in the case of Comte Badoz Reserva.

On the basis of the water content and physical measurements in accordance with the classification criteria, cheeses [42] were divided into four groups: soft, semi-hard, hard, and very hard (Table 1). Table 1 contains information relating to the type of milk used to produce them.

Table 2 presents the results of the content of fatty acid groups in conversion to 100 g of the fat fraction. In the Appendix A, the results of the identified fatty acids in conversion to 100 g of the fat fraction have been included, and additionally the content of groups of fatty acids and identified fatty acids to 100 g of relevant cheese has been converted.

The content of SFA in the tested cheeses was significantly dependent on the type of cheese (*p* < 0.05) and ranged from ca. 55 g 100 g^−1^ in the case of Petit Brillat Savarin to ca. 67 g 100 g^−1^ in Roquefort Papillon (Table 2). SCSFA and BCFA had a significant share in the saturated fatty acid fraction, the largest of which was always butyric acid (C4: 0) and isomers of iso and anteiso acids C13:0, C14:0, C15:0, C17:0, and C18:0 (Appendix A). The SCSFA content was significantly dependent on the type of cheese (*p* < 0.05) and their contents ranged from approx. 11 g 100 g^−1^ in the case of Gorgonzola Cremoso cheese up to approx. 24 g 100 g^−1^ for Papillon Roquefort.

Palmitic acid (C16:0), stearic acid (C18:0), and pentadecanoic acid (C15:0) constituted a significant rest of fatty acids complementing the differentiated profile of the saturated fatty acids of the tested cheeses (Appendix A).

Similarly, in the case of isomers of iso and anteiso saturated fatty acids, their contents have been significantly dependent on the type of cheese (*p* < 0.05). The lowest contents of these isomers were found in Gorgonzola Intenso cheese (approx. 1.5 g 100 g^−1^), and the largest concentrations (approx. 2.9 g 100 g^−1^) in Oscypek, a mountain cheese obtained from sheep’s milk (Table 2). In all analysed cheeses, isomers of acid C15:0 (anteiso C15:0), C16:0 (anteiso C16:0), and C17:0 (iso C17:0) occurred in the greatest concentration (Appendix A). The content of MUFA significantly depended on the type of cheese (*p* < 0.05), and their contents changed even from approx. 15 g 100 g^−1^ (Roquefort Papillon) up to approx. 23 g 100 g^−1^ in the case of Fourmed’Ambert.

The content of MUFA in the tested cheeses mainly depended on the concentration of oleic acid (C18:0), but the profile of MUFA was strongly differentiated by isomers of acid C18:1 in particular, *cis*-11, *cis* 12, and *cis*-14, as well as palmitoleic acid (C16:1 n -7) and other FA occurring in low concentrations (Appendix A). The content of PUFA was from approx. 1.9 g 100 g^−1^ (Pon’t l’evegue) to approx. 4.3 g 100 g^−1^ (Crottin de Chavignol). In the profile of PUFA, the significant share had fatty acids from the family of n-6 and n-3, from which d respectively linoleic acid (C18: 2 n-6) and linolenic acid (C18:3 n-3) occurred in the greatest concentration (Appendix A).

The ratio of n-6 to n-3 fatty acids of the examined cheeses is very advantageous from a nutritional point of view and, for the significant majority of the cheeses, it was within the limits 1:1–4:1 (Appendix A). The PUFA profile of the tested cheeses is very different to what is associated with the occurrence of multiple isomers of C18:2 and C18:3 (Appendix A). A very important share in the PUFA profile constitutes the CLA, the content of which depends strongly on the type of cheese (*p* < 0.05), and in many instances it constitutes a significant share (>50%) of PUFA content.

The highest concentration of CLA has been attributed to Oscypek cheese, both smoked and unsmoked (Polish mountain cheese produced from sheep’s milk, the percentage is greater than 60%)—CLA content was at a level of approx. 2.8 g 100 g^−1^ while the lowest content of CLA was observed in the case of Parmigiano Reggiano (approx. 0.5 g 100 g^−1^). The most important CLA isomers in cheese were CLA *c*9, *t*11 CLA, and *t*9, *c*11, however, a number of other conjugated linoleic acid isomers *t*10, *c*12 have been identified: *c*8, *c*10; *c*9*c*11; *t*12, *t*14; but also at low concentrations *c*11, *t*13; *c*10, *c*12; *c*11, *c*13, and *t*11, *t*13 (Appendix A).

In the fat fraction of the cheeses tested, the significantly differentiated content of *trans* isomers of both monounsaturated and polyunsaturated fatty acids (*p* < 0.05) has been stated. The lowest concentration of *trans*-MUFA was found in cheese Picodon Carte Noire (approx. 1.8 g 100 g^−1^), while the largest in both smoked and non-smoked was Oscypek cheese (approx. 5.9 g 100 g^−1^). The most important *trans*-MUFA isomer determining their content in the fat fraction, and then in cheeses, was C18:1 *t*11 (vaccenic acid), but the profile of *trans*-MUFA is much more diverse in particular with isomers: *t*13 and *t*14; *t*12, *t*9 and *t*8; in addition, at low concentrations, there occur *trans* isomers *t*6, *t*7, and *t*10 of the acid C18:1. We have to underline that, based on scientific controlled intervention data collected by experts of the European Food Safety Authority, studies show that consumption of diets containing *trans*-FA has adverse effects on blood lipids that predict an increase in CHD risk compared with the consumption of diets containing *cis*-monounsaturated fatty acids or *cis*-polyunsaturated fatty acids and that the effect is dose-dependent. Prospective cohort studies showed a consistent association between higher intakes of *trans*-FA and increased risk of CHD. The consistency of the evidence from these two types of studies provides strong support for the conclusion that *trans*-FA intake has a dose-dependent linear effect that increases the risk of CHD as compared to the intake of other fatty acids in the diet. We have some expectations of this rule. One of the most important *trans*-FA found in dairy products fat is vaccenic acid. This acid belongs to the trans-MUFA group. In studies on animals, the hypolipidemic action of C18 11 and *t*11 was indicated. Bassett et al. (2010) [19] and Gómez-Cortés et al. (2018) [20] indicated the reduction of the concentration of triglycerides and LDL in the organism of mice consuming feed containing 15% of butter with a 1.5% share of vaccenic acid. In recent studies by Jacome-Sosa et al. (2014) [21], the authors indicated new mechanisms by means of which vaccenic acid may affect the metabolism of lipids, energy use, and fat tissue distribution in the body. A very important function of vaccenic acid, being the subject of numerous scientific studies, is the inhibition of the development of tumor cells [20,22].

Similar relationships were found in the case of *trans*-PUFA isomers (Table 2) as most of their concentration was observed in the case of sheep cheeses Oscypek both smoked and unsmoked (approx. 3.0 g 100 g^−1^), with the lowest for Picodon Carte Noire, Fromage de chevre au lait cru (Chevre Ronde) and Roquefort Papillon (approx. 1.0 g 100 g^−1^). In addition to CLA isomers, the acids C18:2 *t*9, *t*12; *t*9, *c13,* and *t*9, *c*12 represent a significant share in the profile of polyunsaturated fatty acids with *trans* configuration. However, their profile is much more diverse, which is related to the occurrence of other *trans* isomers of both C18:2 and C18:3.

The tests of cheese conducted in terms of fatty acid content demonstrated that the original commercial yellow and blue cheeses are characterized by a very varied content of fatty acids, but also by their rich profile. Samples of cheeses were subjected to multivariate statistical analysis in order to find similarities and differences between the tested cheeses and to determine what factors influence the differentiation of their fatty acid composition, especially those that are valuable from the health-promoting standpoint.

The results of the hierarchical analysis revealed seven clusterings below the distance of binding, which is 18. The binding distance, 18, was recognized as the critical value because this value starts the fast agglomeration of elements (it was determined based on the course of the agglomeration process, which is shown in Figure 1). This is one of many approaches which allow to determine the end of the agglomeration process and to determine in a quick way the number of clusters. The importance of the clusters has been determined using a one-way analysis of variance with α = 0.05, using the LSD test.

It has been found that six of the seven identified clusters are significantly different from one another (*p* < 0.05). Based on the statistical analysis focusing on the clustering, C7 and C3 are not significantly different from each other *p* = 0.052 (Table 3). In the studied samples of cheese, the first group (C1) are sheep cheeses Oscypek. These cheeses are characterized by the highest content of CLA, of which the average content is 2.7 g 100 g^−1^. In addition, they contain the most isomers of *trans*-FA (both MUFA and PUFA). These cheeses are characterized by a high content of short- and medium-chain fatty acids (approx. 17.0 g 100 g^−1^) and the highest content of branched-chain fatty acids (iso and anteiso over 2.7 g 100 g^−1^).

The second group similar in terms of the content of fatty acids are cheeses in clustering C2. These are long-ripened hard and semi-hard cheeses obtained from cow milk. They are characterized by the lowest content of short- and medium-chain fatty acids, but a relatively high content of stereoisomers of saturated fatty acids (iso and anteiso approx. 2.2 g 100 g^−1^) and *trans*-isomers of the CLA. The cheeses in this group are characterized by a higher share of fatty acids of the n-6 in relation to the n-3 family, compared with group C1. The third group of cheeses are those contained in clusters C3 and C7. In this group are mostly soft and semi-hard cheeses with an overgrowth or growth of mold.

Cheeses in this group are characterized by the lowest content of CLA (<0.6 g 100 g^−1^) including a relatively low content of *trans* isomers of monounsaturated and polyunsaturated fatty acids. The content of stereoisomers of saturated fatty acids in this group of cheeses, as well as of MUFA and PUFA, is the lowest. The content of short- and medium-chain fatty acids in this group ranges at the average level relative to the other cheeses in the other groups (Table 3). The fourth group of cheeses are soft cheeses with a growth of mold obtained from goat milk.

A characteristic feature of these cheeses is the high content of approx. 20 g 100 g^−1^ of short- and medium-chain fatty acids, the highest content of polyunsaturated fatty acids (approx. 3.4 g 100 g^−1^) including the highest acid content from the n-6 (over 2.5 g 100 g^−1^) and n-3 (over 100 g 0.8 g^−1^) family, characterized by the average content of CLA in relation to cheeses in other groups. The fifth group of mold cheeses are soft cheeses and long-ripened hard cheeses (yellow) obtained from cow milk. They are characterized by a low content of stereoisomers of saturated fatty acids, MUFA, PUFA, and the average content of the *trans* isomers, including CLA.

The last group are cheeses in group 6, which consists of both soft and hard blue cheeses obtained from goat or sheep milk. The most important feature of this group of cheeses is the greatest content of saturated fatty acids (over 63.0 g 100 g^−1^) and (over 22.0 g 100 g^−1^) BCFA. Other acids may be considered as occurring at an average level compared to cheeses with other groups.

## 4. Discussion

The conducted studies showed significant differences in the composition of the fatty acids of cheeses obtained from sheep, cow, and goat milk. Moreover, the sort of cheese (soft–very hard) and its type (blue, yellow) also significantly affects the content and profile of fatty acids. Figure 2 shows the results of the classification tree analysis, the purpose of which was to determine which of the above-mentioned factors had the greatest impact on the profile and content of fatty acids in the tested cheeses. Generally, it can be stated that all the factors affect the profile and content of fatty acids in a very significant way, however, the sort of cheese and its type have a great impact on their content.

The influence of the sort and type of cheese on the content of FA plays a significant role because the technological process and the process of cheese ripening are connected with this.

The obtained data are confirmed in the literature, where the influence of the technological process, starter cultures (bacteria, mold), and cheese maturation on the profile and content of FA is repeatedly emphasized [43,44,45].

However, from the point of view of the profile and content of fatty acids, the type of milk from which the cheese has been obtained appears to be a key aspect determining the cheese’s nutritional value.

The use of multivariate statistical analysis has clearly shown that the profile and content of the fatty acids in cheeses depend on the type of milk from which the cheese was obtained. The first group characterized by the highest content of a majority of fatty acid groups were mountain sheep cheeses (Oscypek). Referring to the results obtained from the literature, it is to be concluded that the tested Oscypek were characterized by a very high content of CLA, *trans*-PUFA, vaccenic acid, fatty acids of the n-3 family (mainly α-linolenic acid—ALA), MUFA, BCFA, and SCSFA.

Based on literature data, the fatty acid content in sheep cheeses varies within wide limits: CLA from 0.04 to 3.8 g 100 g^−1^ [43,46,47,48,49,50], *trans*-PUFA od 0.1 do 3.8 g 100 g^−1^ [43,49,50], vaccenic acid from 0.6 to 10.41 g 100 g^−1^ [50,51], fatty acids of the n-3 family from 0.5 to 3.2 g 100 g^−1^ [46,47,48,49,50], MUFA from 18.0 to 58.6 g 100 g^−1^ [48,49,51], BCFA from 0.5 to 2.8 g 100 g^−1^ [49], and SCSFA from 10.5 to 24.0 g 100 g^−1^ [50].

The results obtained within the first group of cheeses are convergent with the literature data, however, the Oscypek cheeses and mountain cheeses obtained from cow milk are characterized by a very high content of health-promoting fatty acids.

The second, third, fifth, and seventh group are cheeses obtained exclusively from cow milk. Referring the contents of individual groups of fatty acids to the literature data, their content is very differentiated: SFA ranges from 53.6 to 75.4 g 100 g^−1^ [41,46,50], SCSFA from 0.09 to 27.45 g 100 g^−1^ [41,46,50], BCFA from 0.02 to 3.0 g 100 g^−1^ [41], MUFA from 18.0 to 30.2 g 100 g^−1^ [41,46,50,51,52], n-3 fatty acids from 0,2 do 0.6 g 100 g^−1^ [41,46,50,52], *trans*-MUFA (mainly vaccenic acid) from 0.23 to 1.4 g 100 g^−1^ [46,50,51], and CLA from 0.29 evet up to 6.1 g 100 g^−1^; however, such high values result from the addition of CLA as a functional component to the final product [46,50].

The obtained data on the content of a group of fatty acids and individual fatty acids are consistent with the literature; however, they are characterized by a great diversity of profile and content. The fourth and sixth groups are goat cheeses.

The fatty acid content in goat cheeses based on literature ranges as follows: SFA from 58 to 74 g 100 g^−1^ [46,53,54,55,56], SCSFA from 11.2 to 37.0 g 100 g^−1^ [46,53,54,55,56], BCFA from 1.8 to 2.0 g 100 g^−1^ [54], MUFA from 19.7 to 31.4 g 100 g^−1^ [46,53,54,55,56] PUFA [46,53,54,55], n-3 from 0.3 to 1.3 g 100 g^−1^ [46,53,54,56], n-6 from 1.4 to 7.0 g 100 g^−1^ [53,54], *trans*-MUFA from 0.2 to 4.9 g 100 g^−1^ [46,54,55,56] *trans*-PUFA from 0.1 to 2.6 g 100 g^−1^ [56] and CLA from 0.1 to 1.0 g 100 g^−1^ [46,53,54,55,56].

The results are consistent with the literature data. The content of SFA in the tested goat cheeses is at a fairly low level, while the content of other groups of fatty acids is consistent with the literature data. The low content of SFA can be explained by the way the animals are fed, that is, the introduction of fresh grass and plants which are quite common during the spring and summer when the samples of cheese were collected.

Profiles and contents of fatty acids in the studied cheeses differ significantly, mainly in the content of health-promoting fatty acids (SCFA, BCFA, *trans*-MUFA—vaccenic acid and CLA). High contents of health-promoting CLA, vaccenic acid, health-promoting BCFA, and long- and medium-chain fatty acids in the results of the tested cheeses, to a significant extent, from the quality of raw materials—the milk used for their production. The profile and content of fatty acids in milk fat are mainly dependent on the microorganisms living in the rumen [57,58].

For this reason, there are very important factors affecting the population of microorganisms in the rumen, among which the type of feed, the proportion of bulky feed in relation to nutritive feed (ratio F:C) in feeding, the chemical composition of feed, rumen pH, and the concentration of ammonia can be mentioned [49,59]. The most important factor responsible for changes in the profile and composition of fatty acids is the content of NDF (Neutral Detergent Fiber) supplied with the feed.

Furthermore, results of studies indicate that by using nutrition of ruminants without fat supplementation, the ruminal bacteria accumulate energy reserves by increasing the synthesis of different groups of fatty acids [60]. The type of feed with which dairy animals are fed affects both the composition of micro-organisms in the rumen, and the rumen microenvironment [31,57,58,61].

An important nutrition factor affecting the content of health-promoting fatty acids in milk, from which cheeses are obtained, is the ratio of F:C (forage: concentrate ratio). If this ratio is higher, the milk fat contains higher quantities of SCSFA, BCFA, *trans*-MUFA, and CLA [31].

To summarize, it is necessary to state that changes in the profile of fatty acids in milk fat are mainly related to the rumen microflora. Factors that accelerate the growth of amylolytic bacteria and decrease the growth of cellulolytic bacteria are mainly an increase in starch content in the feed and the digestibility of the feed, as well as a reduction of the F:C ratio and the content of NDF [62]. The content and profile of fatty acids are also influenced by the stage of lactation [57,58,63].

It has been shown that the quantity of health-promoting fatty acids is greater in the milk fat created at the early lactation stage. A slight negative correlation between the contents of health-promoting fatty acids and the content of milk fat has also been stated [63,64]. All these factors cause a strong differentiation in the content of various groups of fatty acids in the tested cheeses. The highest contents of health-promoting fatty acids identified in Oscypek sheep cheeses probably result from the fact that the sheep were grazed on natural pastures with access to fresh grass, which stimulated the beneficial development of their rumen microflora which, in turn, directly translated into an increase in the content of health-promoting fatty acids in the milk fat.

However, the content and profile of fatty acids in the remaining examined cheeses indicate that the milk used for cheese production came from animals raised in intensive or semi-intensive (mixed) systems. The long process of maturation of the cheese, not without significance, reduces the share of CLA and other fatty acids in the *trans* configuration. During maturation processes of oxidation, changing of the fatty acid profiles takes place, however, to the benefit of taste and smell qualities. The result of the conducted studies is that, from among the tested cheeses, the mountain sheep cheese Oscypek, both smoked and unsmoked, best fits the model of functional food described by Pintus et al. (2013) [25], Lehnen et al. (2015) [26], and Kawęcka et al. (2020) [45].

Based on their unique chemical composition of the lipid fraction, they may significantly affect the reduction of LDL or stabilize the content of triglycerides in the human organism. The Oscypek, in its composition, contains in conversion to fat over 2.7% of CLA, 4.5% of vaccenic acid, 1.3% of ALA (n-3), 2.7% of BCFA, and approx. 17% of SCFA, which decidedly entitles (lub allows) them to state that they are a rich and valuable source of health-promoting fatty acids that can positively affect human health [25,26,45,48].

## 5. Conclusions

Currently, the composition of milk fat and fatty acids functions is newly redefined. Until recently, it has been believed that they are a source of energy accumulated in the fatty tissue and are required to build cell membranes. At present, their role as components demonstrating health-promoting interaction in the human body is emphasized. By analysing the results obtained from the fatty acids content in the samples of cheeses, it can be stated that the content and profile of fatty acids are highly differentiated. The content of fatty acids in the examined cheeses was highly differentiated and depended on the sort and type of cheese. The content of fatty acid groups in milk fat varied within the limits: SFA, 55.2–67.2%; SCSFA, 10.9–23.4%; BCFA, 1.6–2.9%; MUFA, 15.2–23.4%; PUFA, 1.9–4.3%; *trans*-MUFA, 1.8–6.0%; and CLA, 1.0–3.1%. From among the examined cheeses, the seasonal sheep cheeses (“oscypek“) and mountain cow cheeses were characterized by the highest content of health-promoting fatty acids. The content of health-promoting fatty acids in the fat fraction of these cheeses was CLA 2.1–3.1%, *trans*-MUFA 3.5–6%, BCFA 2.7–2.9%, and SCSFA 12–18%. However, in terms of the content of health-promoting fatty acids, the regional sheep cheeses are characterized by a chemical composition beneficial for human health. Based on the works of Lehnen et al. (2015) [26], Pintus et al. (2013) [25], Wilms et al. (2022) [57], and according to Bard et al. (2020) [36], these cheeses may potentially, after introduction into a daily diet, reduce the concentration of LDL and triacylglycerol fraction in the human organism.

Such an advantageous fatty acid composition in the Oscypek cheeses results from them being obtained only periodically, approximately from May to September, when it is possible for the sheep to graze in the mountains where they have access to fresh grass. Extensive breeding of dairy animals is the best way to increase the health-promoting fatty acids in the milk fat of these animals. All tested cheeses were collected in the spring and summer periods in order to eliminate the impact of seasonality. The milk used for the production of these cheeses probably came from dairy animals bred intensively or semi-intensively, which caused the content of health-promoting fatty acids to be lower than in seasonal Oscypek.

## Figures and Tables

**Figure 1 foods-11-01116-f001:**
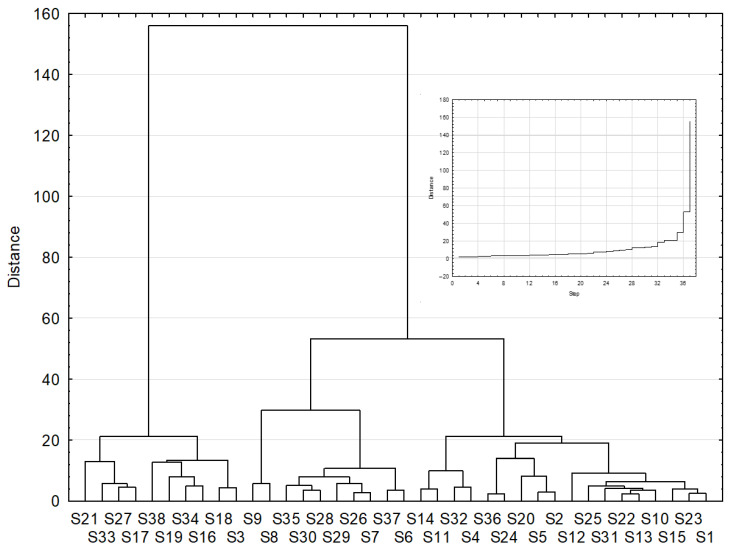
Hierarchical analysis with the process of agglomeration of fatty acid content in studied cheese *n* = 320.

**Figure 2 foods-11-01116-f002:**
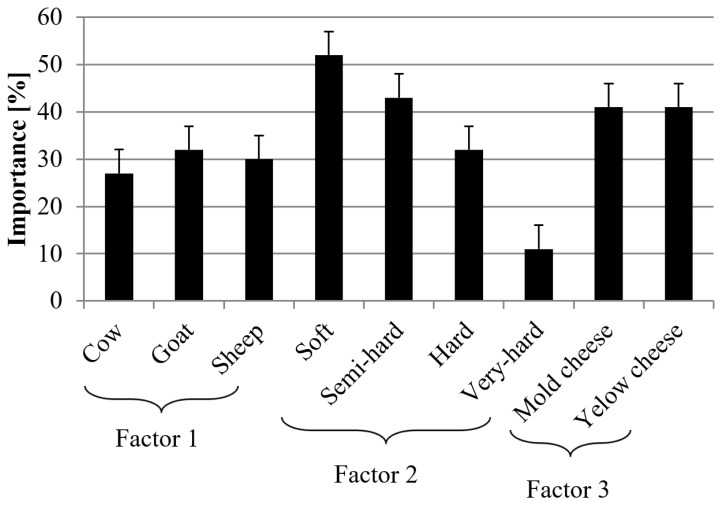
Ranking of priority of factors determining the content of fatty acid in studied commercial cheeses. Ranking on a scale of 0 (low importance) to 100 (high importance), *n* = 320.

**Table 1 foods-11-01116-t001:** Classification, type, and fat content of studied cheeses.

	Fat Content %		Cheese Type	Class. of Cheese
	$\bar{x}$	SD		
S1: Camembert l’aromatique *n* = 8	19.5 ^bc^	0.1	Cow, mould	Soft
S2: Camembert de Caractere, Roi du Chateau *n* = 8	29.2 ^mn^	0.2	Goat, mould	Soft
S3: Buche Fondante *n* = 8	20.5 ^c^	0.4	Goat, mould	Soft
S4: Gorgonzola cremoso *n* = 8	18.3 ^a^	0.9	Cow, mould	Soft
S5: Gorgonzola intense *n* = 8	26.9 ^jk^	0.2	Cow, mould	Soft
S6: Cow mountain cheese non smoked *n* =12	22.5 ^ef^	0.5	Cow, yellow	Semi hard
S7: Cow mountain cheese smoked *n* = 12	25.6 ^hi^	0.5	Cow, yellow	Hard
S8: Sheep mountain cheese “oscypek” non smoked *n* =12	25.2 ^ghi^	0.5	Sheep, yellow	Semi hard
S9: Sheep mountain cheese “oscypek” smoked *n* = 12	27.2 ^k^	0.5	Sheep, yellow	Hard
S10: L’ami du Chambertin *n* = 8	21.9 ^ef^	0.2	Cow, mould	Soft
S11: Brie de Meaux *n* = 8	18.9 ^ab^	0.1	Cow, mould	Soft
S12: Camembert de Normandie *n* = 8	18.8 ^ab^	0.2	Cow, mould	Soft
S13: Epoisses de Bourgogne *n* = 8	21.6 ^d^	0.3	Cow, mould	Soft
S14: Petit Brillat Savarin *n* = 8	30.5 ^pr^	2.2	Cow, mould	Soft
S15: Pon’t l’evegue *n* = 8	22.7 ^f^	0.9	Cow, mould	Soft
S16: Fromage de chevre au lait cru (chevre Ronde) *n* = 8	29.5 ^mnop^	0.4	Goat, mould	Hard
S17: Crottin de Chavignol *n* = 8	25.0 ^ghi^	0.1	Goat, mould	Soft
S18: Le Fleuret *n* = 8	18.0 ^a^	0.2	Goat, mould	Soft
S19: Picodon Carte Noire *n* = 8	19.7 ^bc^	1.5	Goat, mould	Soft
S20: Sainte Maure de Touraine, Hardy Affineur *n* = 8	24.3 ^g^	0.6	Goat, mould	Soft
S21: Selles sur Cher, Appellation d’origine protégée *n* = 8	22.5 ^ef^	0.5	Goat, mould	Soft
S22: English Cheddar Farmhouse *n* = 8	32.0 ^st^	0.3	Cow, yellow	Hard
S23: Emmentaler Francais *n* = 8	24.8 ^gh^	0.4	Cow, yellow	Hard
S24: Raclette Badoz Prest *n* = 8	25.4 ^hi^	0.8	Cow, yellow	Semi hard
S25: Tette de Moine *n* = 8	32.3 ^t^	0.5	Cow, yellow	Hard
S26: Tomme de SavoieYenn *n* = 8	25.9 ^ij^	0.7	Cow, yellow	Hard
S27: Tomme Chevre *n* = 8	25.0 ^ghi^	0.8	Goat, mould	Hard
S28: Appenzeller extra noir *n* = 8	32.0 ^st^	0.6	Cow, yellow	Hard
S29: Beaufort *n* = 8	32.4 ^t^	0.2	Cow, yellow	Hard
S30: Comte Badoz Reserva *n* = 8	33.6 ^u^	0.3	Cow, yellow	Hard
S31: Gruyere sse reserve *n* = 8	30.3 ^opr^	0.2	Cow, yellow	Hard
S32: Parmiggiano Reggiano *n* = 8	27.4 ^k^	0.1	Cow, yellow	Very hard
S33: Manchego Forlasa *n*= 8	29.4 ^mno^	0.3	Sheep, yellow	Hard
S34: Ossau Iraty *n* = 8	31.2 ^rs^	0.4	Sheep, yellow	Hard
S35: Bleu d’Auvergne *n* = 8	25.4 ^hi^	0.4	Cow, mould	Soft
S36: Stilton Colston *n*= 8	29.1 ^mn^	0.7	Cow, mould	Semi hard
S37: Fourme d’Ambert *n* = 8	28.6 ^l^	0.4	Cow, mould	Semi hard
S38: Roquefort Papillon *n* = 8	29.8 ^nop^	0.6	Sheep, mould	Semi hard

a–p, r–u: the mean value with the upper indexes statistically differ (α = 0.05) significantly. Comparative analysis with LSD test, statistical analysis.

**Table 2 foods-11-01116-t002:** Group of fatty acid content in commercial studied cheese samples [g 100 g^−1^].

	SFA		SCSFA **		BCFA ***		MUFA		PUFA		n-3		n-6		*trans*-MUFA		*trans*-PUFA		CLA	
	$\bar{x}$	SD *	$\bar{x}$	SD	$\bar{x}$	SD	$\bar{x}$	SD	$\bar{x}$	SD	$\bar{x}$	SD	$\bar{x}$	SD	$\bar{x}$	SD	$\bar{x}$	SD	$\bar{x}$	SD
Camembert l’aromatique	58.92 cdefghi	2.02	12.95 bcde	0.44	2.08 kl	0.06	18.26 efg	0.68	2.04 abc	0.03	0.56 defg	0.04	1.41 bcd	0.04	3.34 lm	0.13	1.81 hijkl	0.04	0.93 l	0.03
Camembert de Caractere, Roi du Chateau	59.25 defghi	0.47	13.58 efg	0.27	1.89 fghij	0.02	20.33 lmno	0.15	2.18 cde	0.03	0.49 bcde	0.00	1.63 ghi	0.02	2.72 gh	0.08	1.58 ef	0.15	0.68 efg	0.01
Buche Fondante	64.37 mn	3.28	22.82 mo	1.18	1.68 abcde	0.09	18.43 efg	0.97	3.35 pr	0.18	0.75 hi	0.04	2.55 wx	0.14	2.48 de	0.13	1.32 bc	0.09	0.68 efg	0.02
Gorgonzola cremoso	57.40 abcdef	0.81	10.97 a	0.20	1.60 ab	0.20	18.33 efg	0.24	2.78 jkl	0.09	0.57 efg	0.06	2.18 rs	0.04	2.65 efg	0.05	1.72 ghi	0.17	0.52 ab	0.01
Gorgonzola intenso	59.11 cdefghi	1.13	13.22 cdefg	0.56	1.55 a	0.03	19.99 klm	0.39	2.95 lmn	0.05	0.46 bc	0.01	2.45 uw	0.05	2.70 fgh	0.06	1.59 efg	0.01	0.54 ab	0.01
Cow mountain cheese non smoked	56.97 abcdef	0.48	12.07 abc	0.21	2.37 op	0.02	22.84 st	0.25	2.78 jkl	0.01	1.05 mn	0.01	1.66 ghij	0.01	3.57 nop	0.04	1.85 ijkl	0.15	1.09 mn	0.03
Cow mountain cheese smoked	55.51 a	0.26	12.59 bcde	0.49	2.27 mno	0.03	21.17 nopr	0.36	2.91 kl	0.10	1.22 o	0.07	1.62 ghi	0.03	3.47 mno	0.08	2.20 p	0.07	1.09 mn	0.03
Sheep mountain cheese “oscypek” non smoked	56.00 abc	2.82	15.84 h	1.03	2.85 r	0.22	19.58 hijkl	1.57	3.16 o	0.43	1.46 p	0.19	1.55 efg	0.24	5.68 y	0.34	3.11 t	0.17	2.69 u	0.22
Sheep mountain cheese “oscypek” smoked	56.87 abcdef	0.63	18.16 ij	0.83	2.72 r	0.05	18.57 efghi	0.17	2.74 ijk	0.06	1.27 o	0.05	1.32 bc	0.00	5.99 z	0.11	3.03 t	0.04	2.85 w	0.16
L’ami du Chambertin	57.63 abcdefg	3.32	13.01 cdef	0.24	1.93 hij	0.12	17.39 de	1.18	2.33 efg	0.15	0.50 bcdef	0.07	1.77 jkl	0.09	3.01 ij	0.22	1.73 hij	0.10	0.86 jkl	0.05
Brie de Meaux	55.69 ab	1.67	12.27 bcd	0.54	1.65 abc	0.05	18.26 efg	0.56	2.40 fg	0.08	0.52 cdefg	0.06	1.83 lm	0.05	2.89 hi	0.08	1.93 lmn	0.12	0.70 efgh	0.03
Camembert de Normandie	57.75 abcdefg	3.90	14.26 fg	1.30	2.29 no	0.17	16.95 cd	1.01	1.97 ab	0.16	0.73 h	0.07	1.16 a	0.08	4.39 tu	0.25	2.07 nop	0.12	1.40 r	0.09
Epoisses de Bourgogne	58.09 abcdefg	3.90	13.42 defg	1.45	1.77 cdef	0.11	18.89 ghij	1.22	2.50 gh	0.21	0.53 cdefg	0.05	1.92 mn	0.17	3.10 jk	0.11	1.79 hijk	0.09	0.88 kl	0.05
Petit Brillat Savarin	55.23 a	1.87	12.22 abcd	1.16	1.72 bcde	0.03	15.96 abc	0.14	2.34 efg	0.08	0.47 bc	0.07	1.82 klm	0.00	2.48 de	0.01	1.51 de	0.02	0.58 bc	0.01
Pon’t l’evegue	59.01 cdefghi	2.96	13.50 defg	1.22	2.28 mno	0.08	18.67 gh	0.89	1.92 a	0.08	0.52 cdefg	0.01	1.33 b	0.07	5.37 x	0.16	1.78 hijk	0.07	0.92 l	0.05
Fromage de chevre au lait cru (chevre Ronde)	61.22 hijkl	1.49	22.52 mno	1.05	1.80 efgh	0.04	16.57 bcd	0.40	2.75 ijk	0.08	0.56 defg	0.01	2.14 pr	0.07	2.21 c	0.08	1.06 a	0.02	0.65 cde	0.02
Crottin de Chavignol	59.66 efghij	0.43	20.47 k	0.50	1.80 defgh	0.01	18.63 fgh	0.10	4.29 u	0.02	1.64 r	0.03	2.60 x	0.01	2.55 efg	0.02	1.86 jkl	0.02	0.77 hi	0.02
Le Fleuret	64.13 lm	2.32	20.93 kl	1.08	1.66 abcde	0.03	19.03 ghijk	0.62	2.58 hi	0.10	0.34 a	0.02	2.21 rs	0.08	2.51 ef	0.07	1.26 b	0.08	0.56 ab	0.01
Picodon Carte Noire	60.56 ghijk	1.27	19.03 j	0.53	1.95 ijk	0.05	15.90 abc	0.42	2.72 ij	0.06	0.69 h	0.02	1.98 no	0.05	1.85 a	0.05	1.01 a	0.03	0.50 ab	0.02
Sainte Maure de Touraine, Hardy Affineur	58.22 abcdefgh	0.39	17.39 i	0.23	1.63 ab	0.03	19.93 jklm	0.18	2.95 lmn	0.03	0.35 a	0.01	2.57 wx	0.03	2.65 efg	0.03	1.26 b	0.04	0.78 hij	0.03
Selles sur Cher, Appellation d’origine protégée	56.86 abcde	1.26	21.57 klmn	1.00	1.81 efgh	0.05	15.84 ab	0.86	3.94 t	0.01	1.12 n	0.07	2.79 y	0.07	4.07 s	0.13	2.58 s	0.09	1.22 op	0.02
English Cheddar Famhouser	58.16 abcdefgh	0.51	13.02 cdef	0.20	1.69 abcde	0.05	18.75 ghi	0.26	2.04 abc	0.03	0.51 bcdef	0.02	1.46 cde	0.01	2.53 ef	0.06	1.76 hij	0.02	0.74 fghi	0.02
Emmentaler Francais	58.71 bcdefgh	0.48	13.26 cdefg	0.33	1.70 bcde	0.03	18.87 ghij	0.31	2.10 abcd	0.05	0.58 fg	0.02	1.45 cde	0.03	3.84 r	0.14	1.55 ef	0.01	0.79 ij	0.02
Raclette Badoz Prest	61.97 ijklm	0.84	14.35 g	0.28	1.78 cdefg	0.04	19.04 ghijk	0.36	2.12 bcd	0.03	0.43 b	0.02	1.64 ghij	0.04	2.17 c	0.05	1.58 efg	0.07	0.56 ab	0.00
Tette de Moine	59.54 efghi	1.70	12.62 bcde	1.58	2.01 jk	0.04	18.66 gh	0.40	2.40 fg	0.09	0.73 h	0.03	1.59 fgh	0.12	2.67 efg	0.04	1.35 bc	0.01	0.82 ijk	0.01
Tomme de Savoie Yenn	56.37 abcd	1.93	12.74 bcde	0.67	2.24 mno	0.08	21.48 pr	0.70	2.96 lmn	0.13	0.89 jk	0.02	1.99 no	0.11	3.75 pr	0.10	1.94 lmno	0.08	1.29 p	0.04
Tomme Chevre	60.01 fghijk	2.75	21.45 klm	1.06	1.72 bcde	0.09	18.33 efg	0.94	3.16 op	0.15	0.74 hi	0.03	2.37 tu	0.12	2.64 efg	0.12	1.35 bc	0.09	0.73 fghi	0.04
Appenzeller extra noir	56.65 abcde	1.03	13.26 cdefg	0.49	1.91 ghij	0.10	19.74 ijklm	0.49	2.39 fg	0.07	0.88 jk	0.03	1.45 bcde	0.02	4.33 t	0.11	2.02 mno	0.05	1.64 s	0.03
Beaufort	55.66 ab	0.80	12.53 bcde	0.45	2.21 lmn	0.04	20.18 lmn	0.51	3.67 s	0.08	1.58 r	0.04	2.01 no	0.04	4.64 w	0.11	2.37 r	0.08	1.67 s	0.04
Comte Badoz Reserva	56.23 abcd	1.41	13.03 cdef	0.90	2.50 p	0.10	21.38 opr	0.50	2.42 egh	0.06	0.87 j	0.03	1.46 cde	0.05	4.57 uw	0.09	2.07 op	0.12	1.97 t	0.05
Gruyere sse reserve	57.85 abcdefg	1.37	12.79 bcde	0.36	2.45 p	0.11	19.02 ghijk	0.48	2.51 gh	0.07	0.96 kl	0.04	1.48 def	0.04	3.38 lmn	0.09	1.76 hijk	0.02	1.16 no	0.03
Parmiggiano Reggiano	57.39 abcdef	0.41	11.69 ab	0.30	1.78 cdefg	0.03	20.77 mnop	0.31	2.92 klm	0.07	0.60 g	0.04	2.28 st	0.03	2.11 bc	0.03	1.41 cd	0.04	0.49 a	0.01
Manchego Forlasa	60.70 ghijk	0.97	20.39 k	0.27	1.86 fghi	0.05	17.58 def	0.31	3.35 r	0.04	0.48 bcd	0.01	2.83 y	0.04	3.61 op	0.08	1.54 de	0.05	0.66 def	0.03
OssauIrraty	62.82 klm	0.84	22.14 lmno	0.37	1.98 ijk	0.05	16.22 abc	0.24	3.10 mno	0.08	0.97 lm	0.01	2.04 op	0.07	2.86 hi	0.04	1.77 hijk	0.03	0.91 l	0.02
Bleu d’Auvergne	57.14 abcdef	0.51	13.07 cdefg	0.26	2.24 mno	0.04	22.08 rs	0.13	2.50 gh	0.02	0.82 ij	0.01	1.61 ghi	0.02	5.59 y	0.03	1.93 klm	0.02	1.26 p	0.01
Stilton Colston	62.67 jklm	0.92	13.79 efg	0.13	1.63 ab	0.04	18.67 gh	0.20	2.27 def	0.07	0.47 bc	0.01	1.73 ijkl	0.05	2.30 cd	0.06	1.69 fgh	0.13	0.58 bcd	0.02
Fourme d’Ambert	57.93 abcdefg	3.54	12.81 bcde	1.19	2.23 mn	0.11	23.41 t	1.26	2.46 gh	0.11	0.70 h	0.03	1.69 hijk	0.08	3.21 kl	0.18	1.84 ijkl	0.12	1.06 m	0.07
Roquefort Papillon	67.26 n	2.86	23.37 o	1.22	2.15 lm	0.09	15.19 a	0.62	3.11 no	0.14	1.04 lmn	0.05	1.97 no	0.08	1.97 ab	0.13	1.08 a	0.07	0.76 ghi	0.04

* Standard deviation; ** short chain saturated fatty acids; *** sum of iso and anteiso fatty acids (branched-chain fatty acid). a–p, r–u, w–y: the mean value with the upper indexes statistically differ (α = 0.05) significantly. Comparative analysis with LSD test, statistical analysis.

**Table 3 foods-11-01116-t003:** Results of non-hierarchical analysis of group fatty acid content in studied commercial cheese samples, *n* = 320 [g 100 g^−1^].

Clusters (C)		SFA		SCSFA **		BCFA ***		MUFA		PUFA		n-3		n-6		*trans*-MUFA		*trans*-PUFA		CLA	
		$\bar{x}$	SD *	$\bar{x}$	SD	$\bar{x}$	SD	$\bar{x}$	SD	$\bar{x}$	SD	$\bar{x}$	SD	$\bar{x}$	SD	$\bar{x}$	SD	$\bar{x}$	SD	$\bar{x}$	SD
C1	S8: sheep mountain cheese “oscypek” non smoked, S9: sheep mountain cheese “oscypek” smoked,	56.44 a	0.62	17.00 d	1.65	2.79 c	0.09	19.08 b	0.72	2.95 b	0.30	1.37 d	0.13	1.44 a	0.16	5.84 d	0.22	3.07 d	0.06	2.77 d	0.11
C2	S6: Cowmountain cheese non smoked S7: Cow mountain cheese smoked, S26: Tomme de Savoie Yenn, S28: Appenzeller extra noir, S29: Beaufort, S30: Comte Badoz Reserva, S32: Parmiggiano Reggiano, S35: Bleu d’Auvergne, S37: Fourme d’Ambert,	56.65 a	0.80	12.64 a	0.50	2.19 b	0.22	21.45 b	1.19	2.78 b	0.41	0.96 c	0.30	1.75 b	0.28	3.92 c	1.00	1.96 c	0.27	1.28 c	0.43
C3	S24: Raclette Badoz Prest, S36: Stilton Colston	62.32 b	0.50	14.07 c	0.40	1.71 a	0.10	18.85 a	0.26	2.19 a	0.10	0.45 a	0.03	1.69 c	0.06	2.24 a	0.09	1.63 b	0.07	0.57 a	0.01
C4	S17: Crottin de Chavignol, S19: Picodon Carte Noire:, S20: Sainte Maure de Touraine, Hardy Affineur, S21: Selles sur Cher, Appellation d’origine protégée, S27: Tomme Chevre, S33: Manchego Forlasa	59.34 a	1.50	20.05 e	1.59	1.80 a	0.11	17.70 a	1.61	3.40 c	0.60	0.84 c	0.47	2.52 e	0.31	2.89 b	0.80	1.60 b	0.56	0.78 b	0.24
C5	S1: Camembert l’aromatique, S2: Camembert de Caractere, Roi du Chateau, S5: Gorgonzola intenso, S12: Camembert de Normandie, S13: Epoisses de Bourgogne, S15: Pon’t l’evegue, S22: English Cheddar Farmhouse, S23: emmentaler Francais, S25: Tette de Moine, S31: Gruyere sse reserve	58.64 a	0.63	13.26 b	0.47	1.97 a	0.30	18.84 a	0.92	2.26 a	0.32	0.61 b	0.15	1.59 a	0.36	3.40 c	0.90	1.70 b	0.19	0.88 b	0.24
C6	S3: Buche Fondante, S16: Fromage de chevre au lait cru (chevre Ronde), S18: Le fleuret, S34: OssauIrraty, S38: Roquefort Papillon	63.96 b	2.23	22.36 f	0.92	1.85 a	0.21	17.09 a	1.60	2.98 b	0.31	0.73 c	0.29	2.18 d	0.23	2.41 a	0.33	1.30 a	0.29	0.71 b	0.14
C7	S4: Gorgonzola cremoso, S10: L’ami du Chambertin, S11: Brie de Meaux, S14: Petit Brillat Savarin	56.49 a	1.20	12.12 a	0.85	1.73 a	0.15	17.48 a	1.10	2.46 a	0.21	0.52 a	0.04	1.90 c	0.19	2.76 b	0.24	1.72 b	0.17	0.66 a	0.15

* Standard deviation; ** short chain saturated fatty acids; *** sum of iso and anteiso fatty acids (branched-chain fatty acid). a–f: the mean value with the upper indexes statistically differ (α = 0.05) significantly. Comparative analysis with LSD test, statistical analysis.

## Data Availability

Data is contained within the article or Appendix A.

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
