# Peer review of "Content of Health-Promoting Fatty Acids in Commercial Sheep, Cow and Goat Cheeses"

_foods, 2022, doi:10.3390/foods11081116_

Round 1

Reviewer 1 Report

Correction in article is absolutely necessary for improvement.

  1. Introduction is too lengthy. It is necessary to reduce. Negative effect of animal milk fat is necessary to mention introduction. Novelty of investigation is necessary to mention in introduction. Based on literature review, where is knowledge gap, necessary to mention in introduction.
  2. PCA results is needed to summarize the results instead of providing huge data in table. Linear discriminant analysis (LDA) model is required to understand the similarity/ dissimilarity among different samples.
  3. Moisture content is not required in manuscript because however, authors mention the moisture result in manuscript but I cannot find information about it in discussion section.
  4. It is better to include the hardness or texture of cheese and try to make a  correlation with fatty acids.
  5.  Conclusion section is needed to improve with summarize findings and scope for future. 

Author Response

  1. Introduction is too lengthy. It is necessary to reduce. Negative effect of animal milk fat is necessary to mention introduction. Novelty of investigation is necessary to mention in introduction. Based on literature review, where is knowledge gap, necessary to mention in introduction.

The reviewer opinion was implemented in the corrected manuscript.

  1. PCA results is needed to summarize the results instead of providing huge data in table. Linear discriminant analysis (LDA) model is required to understand the similarity/ dissimilarity among different samples.

Thank you for the recommendation to use PCA analysis in our study. I would like to explain that for us the most important was to find the similarity of fatty acid composition of cheese, and try to explain which factor can play on important role in the acids composition. Based on long discussion with a specialist from statistical analysis we decided to use one way ANOVA, hierarchical and non-hierarchical analysis, and classification tree analysis. Now the implementation of PCA analysis is unworkable and we should write a new manuscript to include such data, which lub what is impossible in 10 days for the review process.

  1. Moisture content is not required in manuscript because however, authors mention the moisture result in manuscript but I cannot find information about it in discussion section.

The reviewers opinion was implemented in the corrected manuscript. The information in terms of moisture can be found in Table 1, which includes  the classification, type, moisture and fat content of the studied cheeses

  1. It is better to include the hardness or texture of cheese and try to make a correlation with fatty acids.

In our study we did not make any textural experiments. In the future I am sure that I will use the recommendation of the reviewer and I will try to find a correlation between fatty acid composition and texture.

  1. Conclusion section is needed to improve with summarize findings and scope for future.

The reviewers opinion was implemented in the corrected manuscript.

Reviewer 2 Report

The present work focuses on determining more than 300 samples of different market original sheep cow and goat cheeses, in respect of the content and profile of fatty acids (FA) with special emphasis on health-promoting FA. I must indicate that the originality of the manuscript is poor since there are some studies related to the comparative study of the fatty acid profiles in commercial sheep, cow, and goat cheeses. However, the usefulness of the data is shown well in this manuscript. Although the manuscript was generally understandable, there are some grammatical and syntax errors in the text. Clarity of presentation is fair and scientific merit of the manuscript is worth publishing. Therefore, considerable modifications are needed for the MS according to points that I have indicated. Some major points have been given below: - I suggest English editing from a native English speaker in order to avoid speeling mistakes, grammatical errors, confused and/or ambigues sentences. Abstract The abstract of the manuscript is not well documented to represent the whole research objectives. Abstract should be rewritten again because it not represents the paper sufficiently. Abbreviations used in whole manuscript have to be defined firstly and then their abbreviations have to be used. Please give more numerical data regarding results of the study. Introduction The objectives of the research are not well documented to represent the whole research so please reconstruct the objectives of the paper at the end of introduction section. Materials and Methods All the methods used for the manuscript seem to be fine. Please indicate the city and country for the chemicals and instrument used. Please give reference for all analysis. Please give trade and models of all used chemicals and instruments. Results and Discussion Please give more references related to similar results as stated by authors. Authors should extend the discussion section by giving or comparing the results of the present and previous studies. Discussion of the paper is very weak and has to be modified using with recent references and some excessive explanations have to be removed. Conclusion The conclusion is not suitable since all result are summarised in this section. I think that a brief conclusion should be given what have done in the current investigation. It makes the reader easily follow the results of the research. References Please check the consistency of the references for the format of the journal

Author Response

The present work focuses on determining more than 300 samples of different market original sheep cow and goat cheeses, in respect of the content and profile of fatty acids (FA) with special emphasis on health-promoting FA. I must indicate that the originality of the manuscript is poor since there are some studies related to the comparative study of the fatty acid profiles in commercial sheep, cow, and goat cheeses. However, the usefulness of the data is shown well in this manuscript. Although the manuscript was generally understandable, there are some grammatical and syntax errors in the text.

Clarity of presentation is fair and scientific merit of the manuscript is worth publishing. Therefore, considerable modifications are needed for the MS according to points that I have indicated. Some major points have been given below: - I suggest English editing from a native English speaker in order to avoid speeling mistakes, grammatical errors, confused and/or ambigues sentences.

The reviewers opinion was implemented in the corrected manuscript.

Abstract The abstract of the manuscript is not well documented to represent the whole research objectives. Abstract should be rewritten again because it not represents the paper sufficiently.

The reviewers opinion was implemented in the corrected manuscript.

Abbreviations used in whole manuscript have to be defined firstly and then their abbreviations have to be used. Please give more numerical data regarding results of the study.

The reviewers opinion was implemented in the corrected manuscript.

Introduction The objectives of the research are not well documented to represent the whole research so please reconstruct the objectives of the paper at the end of introduction section.

The reviewers opinion was implemented in the corrected manuscript.

Materials and Methods All the methods used for the manuscript seem to be fine. Please indicate the city and country for the chemicals and instrument used.Please give reference for all analysis. Please give trade and models of all used chemicals and instruments.

The reviewers opinion was implemented in the corrected manuscript.

Results and Discussion Please give more references related to similar results as stated by authors. Authors should extend the discussion section by giving or comparing the results of the present and previous studies.

The reviewers opinion was implemented in the corrected manuscript. Lines from: 395 to 424

Discussion of the paper is very weak and has to be modified using with recent references and some excessive explanations have to be removed.

The reviewers opinion was implemented in the corrected manuscript.

Conclusion The conclusion is not suitable since all result are summarised in this section. I think that a brief conclusion should be given what have done in the current investigation. It makes the reader easily follow the results of the research.

The reviewers opinion was implemented in the corrected manuscript.

References Please check the consistency of the references for the format of the journal

The reviewers opinion was implemented in the corrected manuscript.

Reviewer 3 Report

The ms. “Content of health-promoting fatty acids in commercial sheep, cow and goat cheeses” (Ms. Ref. No. foods-1645403-v1) presents original results regarding the assessment of the content of short and medium chain fatty acids, MUFA, PUFA, CLA and trans fatty acids.

There is a lot of work involved and the ms. is of potential interest.

The topic falls within the aims and scopes of the Foods journal.

However, there are important issues that need to be addressed to improve the presentation of the results.

Major issues:

  1. In the present form, the ms. is biased and misleading on presenting the milk fat (MF) and cheese fat as a source of health promoting fatty acids. Or, it is generally accepted that overall, MF is – on the contrary – highly atherogenic, being constituted mainly of saturated fatty acids (SFA) such as palmitic and stearic, and oleic acid as a MUFA. This aspect should be clearly stated, from the beginning. Only then, the authors should point out that – apart from the atherogenic species – there are other fatty acids, in smaller amounts, which have proved beneficial health effects. From this viewpoint, the ms. should be extensively revised and re-organized. Please revise.
  2. The Introduction should be re-organized to present first the MF composition on the major fatty acids, and afterward the minor species.
  3. Please point out the elements of novelty/originality of the present work.
  4. Line 34: In the present form, this sentence is confusing. Revise according to comment #2, then mention that the minor species occur in less than 1%.
  5. Line 40: The abbreviation is not suggestive to include the medium chain FA. In addition, the literature generally differentiates these fatty acids into two distinct classes: short chain saturated fatty acids (SCFA) and medium chain fatty acids (MCFA). I suggest referring to the two categories independently, not as a single class. I also suggest pointing out that the significant amounts of SCFA and MCFA is characteristic to milk fats and a discriminant factor to differentiate milk fats from other fats and oils. Please revise. This also implies re-calculation of the corresponding classes in tables.
  6. Lines 58-59: I do not agree: MUFA are neither beneficial, not detrimental. You should mention this and you can only discuss the situation of vaccenic acid. Here, you should mention that vaccenic acid belongs to MUFA, but is also a trans FA (TFA). In the current form, from the Abstract and other parts of the ms., it results that TFA are health promoting species, which is wrong. The authors should mention that TFA increase the risk of cardiovascular disease, by changing the plasma lipoprotein profile in the favor of low-density lipoproteins (LDL), therefore it was suggested that the human intake should be less than 2% as reported to the overall fatty acids (Mihalache et al., “Thermal Formation of trans Fatty Acids in Romanian Vegetable Oils Monitored by GC-MS and FT-IR Techniques”, Revista de Chimie, 63(10), 2012, pp. 984-988). Only after that (in this context), the authors should point out the specific situation of vaccenic acid as an exception from the rest of the TFA and include lines 60-67. This way, acknowledging both detrimental and benefic qualities, the ms. will not be biased. Please revise.
  7. Lines 70-71: I do not understand what the authors mean. Please rephrase.
  8. Lines 103-104: I agree and this aspect should be elaborated. I suggest referring to https://doi.org/10.3168/jds.2021-21411 and https://doi.org/10.3168/jds.2021-20728 to showcase the altered milk fat composition in favor of the health promoting species discussed in your manuscript. You should also mention the second reference when discussing CLA species and vaccenic acid. In addition, it is worth mentioning that - given the low content of PUFA in milk fat - recent preoccupations regard improving the fatty acid profile of dairy products through addition of exogeneous oils (see, for example: https://doi.org/10.1016/j.lwt.2021.112793 and  https://doi.org/10.1016/j.lwt.2022.113105)
  9. Lines 116-119: Again, misleading. Health promoting FA account for approx. 15%, while the rest are atherogenic species. Therefore, MF can never be considered as a healthy fat (or functional food). Please revise.
  10. Sub-section 2.5: Please include sample concentration for GC. How was the FAME quantification performed?
  11. Line 150: I do not understand. What blends? Please revise.
  12. Sub-section 2.6: How many replicates for each sample?
  13. Water content is not significant for the investigated issue. It only makes the ms. longer in a sterile manner. Please remove any data and comments regarding water content.
  14. Table 1 and R&D sections: Is it ok to give brand names? Please ask for the Editor’s opinion. Lines 187-190: I suggest expressing the mean values for cheese type.
  15. Tables 2 and 3: The presentation of data is odd. I suggest presenting data as mean values ± sd, and the significant differences letters as superscript. Also, state the meaning of these letters (are they Duncan’s letters, Tuckey test, or what) as a table footnote.
  16. Line 209: SFA was used in tables, but the abbreviation was explained only in line 209; it should be explained previously.
  17. Lines 211-213: I do not understand. Please rephrase.
  18. Lines 218-220: These species are atherogenic, therefore, not healthy. I suggest eliminating this discussion. Focus on the healthy categories!
  19. Lines 232-239: These results should be compared to other results reported in the literature. I suggest comparing the cheese composition. In addition, I also recommend these relevant references: https://doi.org/10.3168/jds.2021-20961.
  20. Regarding the n-3 and n-6 FA: the authors should point out that n-3 and n-6 FA are found in low amounts in milk fat (generally, less than 2%). However, these amounts may vary, depending on various factors, including stage of lactation (https://doi.org/10.3168/jds.2021-20539 and https://doi.org/10.3168/jds.2021-20880). In addition, the authors should elaborate on the importance of linoleic acid (n-6 species), as an essential fatty acid, being starting point for the biosynthesis of a series of ω-6 fatty acids, eventually leading to arachidonic acid C20:4.
  21. In the current form, the References section include many ol or very old references, with a negative impact on the argumenting the actuality of the investigated topic. You need to point out the adequacy of your paper in the current research landscape. And recent papers can advocate for the actuality of the theme.

Minor issues:

  1. Please wright cis/trans with Italics throughout the ms; same for all the Latin names (g. for microorganisms) or Latin words (e.g. de novo line 37, vernix caseosa, line 92). Do it throughout the ms.
  2. Line 40: saturated not with Capital S.
  3. Line 60: Please revise lipid number.
  4. Line 89: Define KT.
  5. I recommend naming Section 5 as Conclusions.
  6. I suggest adding a List of Abbreviations.
  7. Please revise English throughout the ms. There are several syntax/grammar/typos which need to be fixed. Revise verb tenses.

Given the completed score sheet and the comments above, after careful evaluation, the ms. “Content of health-promoting fatty acids in commercial sheep, cow and goat cheeses” (Ms. Ref. No. foods-1645403-v1) needs Major Revision according to the comments.

Author Response

  1. In the present form, the ms. is biased and misleading on presenting the milk fat (MF) and cheese fat as a source of health promoting fatty acids. Or, it is generally accepted that overall, MF is – on the contrary – highly atherogenic, being constituted mainly of saturated fatty acids (SFA) such as palmitic and stearic, and oleic acid as a MUFA. This aspect should be clearly stated, from the beginning. Only then, the authors should point out that – apart from the atherogenic species – there are other fatty acids, in smaller amounts, which have proved beneficial health effects. From this viewpoint, the ms. should be extensively revised and re-organized. Please revise.

The reviewer opinion was implemented in the corrected manuscript. In the ms we added some part in introduction in terms of negative aspects of milk fat.

  1. The Introduction should be re-organized to present first the MF composition on the major fatty acids, and afterward the minor species.

The reviewer opinion was implemented in the corrected manuscript. In the ms we added some part in introduction in terms of negative/positive aspects of milk fat and major fatty acids.

  1. Please point out the elements of novelty/originality of the present work.

The reviewer opinion was implemented in the corrected manuscript.

  1. Line 34: In the present form, this sentence is confusing. Revise according to comment #2, then mention that the minor species occur in less than 1%.

The reviewer opinion was implemented in the corrected manuscript.

  1. Line 40: The abbreviation is not suggestive to include the medium chain FA. In addition, the literature generally differentiates these fatty acids into two distinct classes: short chain saturated fatty acids (SCFA) and medium chain fatty acids (MCFA). I suggest referring to the two categories independently, not as a single class. I also suggest pointing out that the significant amounts of SCFA and MCFA is characteristic to milk fats and a discriminant factor to differentiate milk fats from other fats and oils. Please revise. This also implies re-calculation of the corresponding classes in tables.

Short chain saturated fatty acids and medium chain fatty acids are not well defined in literature and we do not have a strict definition of it. We decided to connect together SCFA and MCFA because the acids are typical for milk products and play similar biological role in human body. In supplementary matterials we showed the whole identified fatty acids and if somebody need to do meta-analysis can use the results to calculate of Medium chain fatty acids. In our tables we show SCFA (both SCFA and MCFA), BCFA, trans-MUFA and CLA fatty acids, which are typical for dairy products fat.

  1. Lines 58-59: I do not agree: MUFA are neither beneficial, not detrimental. You should mention this and you can only discuss the situation of vaccenic acid. Here, you should mention that vaccenic acid belongs to MUFA, but is also a trans FA (TFA). In the current form, from the Abstract and other parts of the ms., it results that TFA are health promoting species, which is wrong. The authors should mention that TFA increase the risk of cardiovascular disease, by changing the plasma lipoprotein profile in the favor of low-density lipoproteins (LDL), therefore it was suggested that the human intake should be less than 2% as reported to the overall fatty acids (Mihalache et al., “Thermal Formation of trans Fatty Acids in Romanian Vegetable Oils Monitored by GC-MS and FT-IR Techniques”, Revista de Chimie, 63(10), 2012, pp. 984-988). Only after that (in this context), the authors should point out the specific situation of vaccenic acid as an exception from the rest of the TFA and include lines 60-67. This way, acknowledging both detrimental and benefic qualities, the ms. will not be biased. Please revise.

The reviewer opinion was implemented in the corrected manuscript.

  1. Lines 70-71: I do not understand what the authors mean. Please rephrase.

The reviewer opinion was implemented in the corrected manuscript.

  1. Lines 103-104: I agree and this aspect should be elaborated. I suggest referring to https://doi.org/10.3168/jds.2021-21411 and https://doi.org/10.3168/jds.2021-20728 to showcase the altered milk fat composition in favor of the health promoting species discussed in your manuscript. You should also mention the second reference when discussing CLA species and vaccenic acid. In addition, it is worth mentioning that - given the low content of PUFA in milk fat - recent preoccupations regard improving the fatty acid profile of dairy products through addition of exogeneous oils (see, for example: https://doi.org/10.1016/j.lwt.2021.112793 and https://doi.org/10.1016/j.lwt.2022.113105)

The reviewer opinion was implemented in the corrected manuscript.

  1. Lines 116-119: Again, misleading. Health promoting FA account for approx. 15%, while the rest are atherogenic species. Therefore, MF can never be considered as a healthy fat (or functional food). Please revise.

In this paragraph we put emphasis to discuss results of Pintus et al. (2013) and Wilms at al. (2022) in comparison with COMMISSION REGULATION (EU) No 432/2012 of 16 May 2012 establishing a list of permitted health claims made on foods, other than those referring to the reduction of disease risk and to children’s development and health. We corrected the aim of the article.

  1. Sub-section 2.5: Please include sample concentration for GC. How was the FAME quantification performed?

FA methyl esters were prepared and calculated according to AOAC Official Method 2012.13 protocol and analysed by via gas chromatography.

  1. Line 150: I do not understand. What blends? Please revise.

It was corrected

  1. Sub-section 2.6: How many replicates for each sample?

It was corrected, the n = 8

  1. Water content is not significant for the investigated issue. It only makes the ms. longer in a sterile manner. Please remove any data and comments regarding water content.

Water content was important in discussion section. Moreover water content is important to show what kind of cheese we have for example hard, soft or medium hart. It play important role during ripening of cheese and the process was discussed in the manuscript. In our opinion showing this data is advantage than disadvantage.  If reviewer still has a opinion to remove this information we will do it.

  1. Table 1 and R&D sections: Is it ok to give brand names? Please ask for the Editor’s Lines 187-190: I suggest expressing the mean values for cheese type.

We do not have any conflict of interest. For readers it is much better to show the true names of studied samples

  1. Tables 2 and 3: The presentation of data is odd. I suggest presenting data as mean values ± sd, and the significant differences letters as superscript. Also, state the meaning of these letters (are they Duncan’s letters, Tuckey test, or what) as a table footnote.

The reviewer opinion was implemented in the corrected manuscript.

  1. Line 209: SFA was used in tables, but the abbreviation was explained only in line 209; it should be explained previously.

We added abbreviation section in the beginning of the ms

  1. Lines 211-213: I do not understand. Please rephrase.

It was corrected

  1. Lines 218-220: These species are atherogenic, therefore, not healthy. I suggest eliminating this discussion. Focus on the healthy categories!

It was corrected

  1. Lines 232-239: These results should be compared to other results reported in the literature. I suggest comparing the cheese composition. In addition, I also recommend these relevant references: https://doi.org/10.3168/jds.2021-20961.

It was corrected in discussion section

  1. Regarding the n-3 and n-6 FA: the authors should point out that n-3 and n-6 FA are found in low amounts in milk fat (generally, less than 2%). However, these amounts may vary, depending on various factors, including stage of lactation (https://doi.org/10.3168/jds.2021-20539 and https://doi.org/10.3168/jds.2021-20880). In addition, the authors should elaborate on the importance of linoleic acid (n-6 species), as an essential fatty acid, being starting point for the biosynthesis of a series of ω-6 fatty acids, eventually leading to arachidonic acid C20:4.

It is very good information, we discuss a lot of aspect, factors which play important role in fatty acid synthesis. We added the literature position to our manuscript.

  1. In the current form, the References section include many ol or very old references, with a negative impact on the argumenting the actuality of the investigated topic. You need to point out the adequacy of your paper in the current research landscape. And recent papers can advocate for the actuality of the theme.

The reviewer opinion was implemented in the corrected manuscript.

Minor issues:

  1. Please wright cis/trans with Italics throughout the ms; same for all the Latin names (g. for microorganisms) or Latin words (e.g. de novo line 37, vernix caseosa, line 92). Do it throughout the ms.

It was corrected

  1. Line 40: saturated not with Capital S.

It was corrected

  1. Line 60: Please revise lipid number.

It was corrected

  1. Line 89: Define KT.

It was corrected

  1. I recommend naming Section 5 as Conclusions.

It was corrected

  1. I suggest adding a List of Abbreviations.

It was corrected

  1. Please revise English throughout the ms. There are several syntax/grammar/typos which need to be fixed. Revise verb tenses.

It was corrected

Round 2

Reviewer 1 Report

Authors are not able to modify the manuscript based on my suggestion. They replied it is due to shortage of time. Realizing the situation, I recommend to accept this manuscript. Editors can decide based on situation. I do not know that extension of deadline is possible or not..

Author Response

Thank you very much for the Reviewer's opinion

Reviewer 2 Report

The paper has been revised according to the suggestions and criticisms of the reviewers. In this revised version, the paper has improved its quality.

Author Response

(The authors gave the same response as above.)

Reviewer 3 Report

The authors of the revised (revision 1) ms. “Content of health-promoting fatty acids in commercial sheep, cow and goat cheeses" (Ms. Ref. foods-1645403-v2) have addressed the majority of the reviewers’ comments in an appropriate manner. Proper changes were performed on the manuscript. Consequently, the quality of the paper was considerably improved, as compared to its initial submission.

However, there are still a few issues that need to be addressed to polish the ms. before recommending it for publication:

  1. I have carefully studied the authors’ rebuttal to my comment #13 (regarding the water content) and I still consider the results and corresponding discussion do not fit into this paper. The paper aims at discussing the composition of the cheese fat with a focus on the beneficial fatty acid species, the moisture not only does not bring any advance or significant correlation with the investigated species, but also distract the attention from the main topic. So, yes, I maintain my opinion and recommend removing this part. Please revise.
  2. Regarding the discussion on the trans fatty acid species: in the first review round, I have suggested the authors to point out the specificity of vaccenic acid against the trans fatty acids family (see comment #6). In this respect and correlated with the fact that the ms. also investigates the occurrence of other trans fatty acids than vaccenic acid, it is important to mention that TFA increase the risk of cardiovascular disease, by changing the plasma lipoprotein profile in the favor of low-density lipoproteins (LDL), therefore the limitation of its intake in human diet was recommended (see comment #6 from the first review round for the details). The limitation in the diet should be explicitly indicated. Only afterwards the authors should highlight the specificity of vaccenic acid, as an exception of health promoting trans fatty acid. Please revise.
  3. I suggest the authors to specifically indicate in their responses to reviewers’ comments the exact lines where the modifications have been performed. In this way, it would be easier for both editors and reviewers to correlate the comments with the corresponding modifications in the ms., without any confusion. Responses such as: “The reviewer opinion was implemented in the corrected manuscript.”, without indicating where specifically it was addressed in the text, are not helpful at all. Please consider this for the next revision.

Given the completed scoresheet and the comments above, after careful evaluation, my recommendation term is Minor revision according to comments. I look forward to receiving the revised version of the ms.

Kind regards.

Author Response

I have carefully studied the authors’ rebuttal to my comment #13 (regarding the water content) and I still consider the results and corresponding discussion do not fit into this paper. The paper aims at discussing the composition of the cheese fat with a focus on the beneficial fatty acid species, the moisture not only does not bring any advance or significant correlation with the investigated species, but also distract the attention from the main topic. So, yes, I maintain my opinion and recommend removing this part. Please revise.

We decided to agree with Reviewer opinion. We have implemented Reviewer opinion to the manuscript

Regarding the discussion on the trans fatty acid species: in the first review round, I have suggested the authors to point out the specificity of vaccenic acid against the trans fatty acids family (see comment #6). In this respect and correlated with the fact that the ms. also investigates the occurrence of other trans fatty acids than vaccenic acid, it is important to mention that TFA increase the risk of cardiovascular disease, by changing the plasma lipoprotein profile in the favor of low-density lipoproteins (LDL), therefore the limitation of its intake in human diet was recommended (see comment #6 from the first review round for the details). The limitation in the diet should be explicitly indicated. Only afterwards the authors should highlight the specificity of vaccenic acid, as an exception of health promoting trans fatty acid. Please revise.

We agreed with Reviewer opinion and suggestion before (according with firs review) – Reviewer opinion was very crucial and important. We implemented correction in introduction section in terms of MUFA and vaccenic acid because more than one reviewer put attention for this topic. I understand that reviewer has a wish to add strong impact on trans fatty acids. We did it in discussion section. I hope that added part of text will be satisfied for reviewer.

I suggest the authors to specifically indicate in their responses to reviewers’ comments the exact lines where the modifications have been performed. In this way, it would be easier for both editors and reviewers to correlate the comments with the corresponding modifications in the ms., without any confusion. Responses such as: “The reviewer opinion was implemented in the corrected manuscript.”, without indicating where specifically it was addressed in the text, are not helpful at all. Please consider this for the next revision.

The firs Reviewer opinion was implemented in materials and results section: table 1 and lines 162-163, 192-194, 228-233

The second Reviewer opinion was implemented in discussion section lines 302- 320